# The Anti-Cancer Drug Dabrafenib Is a Potent Activator of the Human Pregnane X Receptor

**DOI:** 10.3390/cells9071641

**Published:** 2020-07-08

**Authors:** Nicolas Creusot, Matthieu Gassiot, Elina Alaterre, Barbara Chiavarina, Marina Grimaldi, Abdelhay Boulahtouf, Lucia Toporova, Sabine Gerbal-Chaloin, Martine Daujat-Chavanieu, Alice Matheux, Roger Rahmani, Céline Gongora, Alexandre Evrard, Philippe Pourquier, Patrick Balaguer

**Affiliations:** 1Institut de Recherche en Cancérologie de Montpellier, Inserm U1194, Université de Montpellier, ICM, 34298 Montpellier, France; nicolas.creusot@inrae.fr (N.C.); matthieu.gassiot@gmail.com (M.G.); elina.alaterre@igh.cnrs.fr (E.A.); barbara.chiavarina@inserm.fr (B.C.); marina.dilillo-grimaldi@inserm.fr (M.G.); abdel.boulahtouf@inserm.fr (A.B.); lucia.toporova@inserm.fr (L.T.); alice.matheux@inserm.fr (A.M.); celine.gongora@inserm.fr (C.G.); alexandre.evrard@umontpellier.fr (A.E.); 2IRMB, Université de Montpellier, INSERM, CHU Montpellier, 34090 Montpellier, France; sabine.gerbal-chaloin@inserm.fr (S.G.-C.); martine.daujat@inserm.fr (M.D.-C.); 3INRA UMR 1331 TOXALIM, 06560 Sophia Antipolis, France; roger.rahmani@inrae.fr

**Keywords:** dabrafenib, hPXR, colon and liver cancer cells, proliferation

## Abstract

The human pregnane X receptor (hPXR) is activated by a large set of endogenous and exogenous compounds and plays a critical role in the control of detoxifying enzymes and transporters regulating liver and gastrointestinal drug metabolism and clearance. hPXR is also involved in both the development of multidrug resistance and enhanced cancer cells aggressiveness. Moreover, its unintentional activation by pharmaceutical drugs can mediate drug–drug interactions and cause severe adverse events. In that context, the potential of the anticancer BRAF inhibitor dabrafenib suspected to activate hPXR and the human constitutive androstane receptor (hCAR) has not been thoroughly investigated yet. Using different reporter cellular assays, we demonstrate that dabrafenib can activate hPXR as efficiently as its reference agonist SR12813, whereas it does not activate mouse or zebrafish PXR nor hCAR. We also showed that dabrafenib binds to recombinant hPXR, induces the expression of hPXR responsive genes in colon LS174T-hPXR cancer cells and human hepatocytes and finally increases the proliferation in LS174T-hPXR cells. Our study reveals that by using a panel of different cellular techniques it is possible to improve the assessment of hPXR agonist activity for new developed drugs.

## 1. Introduction

Because cancer patients are usually treated with a combination of medications, a critical issue of the drug development process is the potential existence of drug–drug interactions (DDIs). DDIs can directly impact drugs’ pharmacokinetics and result in reduced efficacy of the treatment or in the occurrence of severe adverse events. In accordance with FDA (Food and Drug Administration) and EMA (European Medicines Agency) guidelines, such a risk can be assessed by evaluating the potential of any new drug to act as inducer of clinically relevant metabolizing enzymes such as cytochrome P450 or UGT enzymes [1].

Since hPXR and receptor hCAR are considered as master regulators of most of these metabolizing enzymes or transporters at the transcriptional level, they could indirectly modulate the activity of various drugs and be responsible for DDIs when they are co-administered [2]. This is particularly true for pharmaceuticals and anticancer drugs such as paclitaxel, flutamide or tamoxifen, which can activate hPXR and hCAR [3,4,5]. Chemoresistance to these agents is often associated with activation of hPXR and hCAR and the expression of their target genes in various cancer cell types such as ovary, prostate, liver, intestinal and colon [6,7,8,9,10,11,12,13]. Other compounds that are routinely co-administrated with anticancer agents such as corticosteroids, anticoagulants, analgesics, antibiotics, antiemetics, anticonvulsants or antiepileptics were also shown to activate hPXR and hCAR [4,14]. Thus, it is critical to take into account the potential of hPXR and hCAR activation in the rational development of anticancer drug combinations to limit DDIs and reduce the risk of adverse events. Although the mechanism is not fully elucidated, hPXR and hCAR may also be involved in the modulation of tumor progression. Indeed, the ectopic expression of hPXR stimulates cell proliferation of HepG2 liver cancer cells [15]. Furthermore, in vitro and in vivo activation of endogenous hPXR in colon cancer cells [16,17] also results in enhanced cell growth.

Dabrafenib is a reversible, highly potent ATP-competitive inhibitor of BRAF serine/threonine kinase involved in the regulation of the mitogen-activated protein kinase (MAPK) pathway. Dabrafenib was developed to specifically target the V600E activating mutation of BRAF (BRAF^V600E^), which is observed in more than 40% of melanomas and probably contributes to the progression of the disease. In vitro studies reported an IC_50_ value of 0.65 nM for BRAF^V600E^ [18].

Dabrafenib is currently approved alone or in combination with the MEK inhibitor trametinib for the treatment of metastatic melanoma [19]. Interestingly, the prescription notice of dabrafenib indicates that it is metabolized by CYP2C8 and CYP3A4, while its metabolites hydroxy-dabrafenib and desmethyl-dabrafenib are CYP3A4 substrates [20]. In addition, dabrafenib is described as an inducer of CYP3A4, CYP2C9, CYP2C8, CYP2C19 and of UGT enzymes that could lead to a decrease in the intracellular concentrations of their respective substrates [20]. While these effects may be due to hPXR- or hCAR-mediated transcriptional effects, the possibility that dabrafenib could directly affect the activity of these nuclear receptors has not been fully investigated yet.

In this work and using a panel of binding, reporter and proliferation cellular assays and RT-qPCR experiments in LS174T-hPXR and primary human hepatocytes, we were able to characterize the PXR activity of dabrafenib.

## 2. Materials and Methods

### 2.1. Reagents and Chemicals

6-(4-Chlorophenyl)imidazo [2,1-b][1,3]thiazole-5-carbaldehyde O-(3,4-dichlorobenzyl) oxime (CITCO), clotrimazol, Pregnenolone 16 alpha-carbonitrile (PCN) and SR12813 were purchased from Sigma-Aldrich (Saint Quentin Fallavier, France). Dabrafenib was purchased from Euromedex (Souffelweyersheim, France). SPA70 was purchased from Axon (Groningen, The Netherlands). Drugs were solubilized in DMSO at 10 mM and stored at −20 °C.

### 2.2. PXR and CAR Reporter Cell Lines

The different reporter cell lines used in this study are summarized in Appendix A. HG5LN GAL4-m/zfPXR cells were obtained as previously reported for HG5LN GAL4-hPXR cells [21,22]. Briefly, Hela cells were stably transfected with the GAL4RE_5_-βGlob-Luc-SVNeo plasmid alone or together with the pSG5-GAL4(DBD)-h/m/zfPXR(LBD)-puro plasmids leading to the HG5LN and HG5LN-h/m/zfPXR cell lines, respectively. HG5LN GAL4-hCAR and GAL4-hCAR-APYLT cells were obtained in the same way.

The LS174T-hPXR reporter cell line was previously described [22]. These cells were obtained through stable co-transfection of LS174T colon cancer cells with the pcDNA3.1-hPXR (1-434)-neomycin plasmid and the XREM-CYP3A4-luciferase-hygromycin reporter plasmid. Based on qRT-PCR experiments, the hPXR expression is 5 to 10 times higher in LS174T-hPXR cells compared to that of human hepatocytes or differentiated HepaRG cells (Appendix A). The liver cancer HepG2-hPXR [22] and the prostate cancer 22RV1-hPXR cell lines were obtained using a similar protocol.

Cells were cultured at 37 °C under humidified 5% CO_2_ atmosphere. HG5LN GAL4-h/m/zfPXR cells were cultured in phenol red (DMEM)-F12 medium (Thermofisher, Villebon sur Yvette, France) supplemented with 5% fetal calf serum (FCS), 1% penicillin/streptomycin (100 U/mL), 0.5 µg/mL puromycin and 1 mg/mL G418. LS174T and HepG2-hPXR cells were cultured in phenol red (DMEM)-F12 medium (Thermofisher, Villebon sur Yvette, France) supplemented with 5% fetal calf serum (FCS), 1% penicillin/streptomycin (100 U/mL) and 1 mg/mL G418. For HepG2 cells, culture medium was supplemented with sodium pyruvate (1%), hepes (1%) and non-essential amino acid (1%). 22RV-1 cells were cultured in RPMI medium supplemented with 5% FCS, 1% penicillin/streptomycin (100 U/mL), 1 mg/mL G418 and 0.25 mg/mL hygromycin.

The induction of luciferase activity by PXR ligands was stable for more than 3 months for the different reporter cell lines in culture, which corresponds to 10–15 passages. During these passages, we did not observe significant variability in luciferase induction in both absolute luminescence and fold induction.

### 2.3. Transactivation Assays

Cells were seeded in 96-well, white, opaque, flat bottom plates at 40,000 cells per well in 150 µL of DMEM-F12 without phenol red supplemented with penicillin/streptomycin (1%) and dextran-coated charcoal-treated fetal calf serum (DCC-FCS) (5%) (test medium). Compounds were added 24 h later using automated workstation (Biomek 3000, Beckman Coulter, Villepinte, Paris), and cells were incubated at 37 °C for 16 h. Then, medium was removed, and 50 µL of test medium containing luciferin at 0.3 mM was added per well. Luciferase activity was measured for 2 s in intact living cells after 10 min stabilization using a MicroBetaWallac luminometer (PerkinElmer). Each compound was tested at various concentrations in at least three independent experiments. For each experiment, tests were performed in quadruplicate for each concentration, and data are expressed as mean values with standard deviations. Individual agonist dose–response curves were fitted using the sigmoid dose–response function of GraphPad Prism (version 5.0). The equation used by Graphpad was
Y = Bottom + (Top-Bottom)/(1 + 10^((LogEC50-X) × HillSlope)),
where X is the log of concentration and Y is the response.

EC_50_ (effective concentration for half-maximal luciferase activity) and IC_50_ (half-maximal inhibitory concentration) values were calculated. To analyze significances, we compared individual compound treatments with controls using one-way analysis of variance (ANOVA) using the GraphPad Prism software.

### 2.4. LS174T-hPXR Proliferation Assay (P-SCREEN)

We developed a new proliferation test that we called the P-screen to measure the effects of compounds with PXR activity on the growth of LS174T-hPXR cells. This test uses the same principle as that for E-screen that was used for measuring the effects of estrogens on the proliferation of MCF-7 breast cancer cells [23]. Briefly, LS174T-hPXR cells were seeded in 96-well plates (2000 cells per well) in test medium containing the hPXR antagonist SPA70 [24] at 100 nM for 24 h. This pretreatment with SPA70 was performed in order to reduce the basal proliferation of the cells, thus allowing a more sensitive detection of any PXR agonist activity that would result in growth induction. The medium was then removed, and cells were treated with increasing concentrations of the different compounds for 7 days. Medium was removed, and 0.1 mL of MTT solution (0.5 mg/mL) was added to each well. The MTT-containing medium was removed 4 h after incubation, and DMSO was added to each well. After shaking, the plates were read in absorbance at 540 nm. Results were expressed as percentage of proliferation with respect to the hormone-free control (100%). Data were obtained by dose–response curves plotted as percentage of proliferation as a function of concentrations.

The activation of the proliferation by PXR ligands was stable for more than 3 months for the LS174T-hPXR cell line in culture, which corresponds to 10–15 passages. During these passages, we did not observe significant variability in proliferation induction.

### 2.5. Lanthascreen TR-FRET PXR Competitive Binding Assay

GST-hPXR-LBD (10 nM) was incubated with different concentrations (10–30 μM) of dabrafenib and SR12813 in the presence of Fluormone PXR ligand (40 nM) and Lanthascreen terbium-anti-GST antibody (10 nM). To read a LanthaScreen TR-FRET assay, the fluorimeter (PHERAstar FS; BMG LABTECH) was configured to excite the terbium donor around 340 nm, and to separately read the terbium emission peak that is centered at 490 nm and the fluorescein emission that is centered at 520 nm. Results are expressed as the signal from the fluorescein emission divided by the terbium signal to provide a TR-FRET emission ratio. Fluorescence ratio data were fitted using a sigmoidal dose–response model using GraphPad Prism (GraphPad Software Inc, San Diego, CA, USA).

### 2.6. Isolation and Primary Culture of Human Hepatocytes

Liver samples were obtained from the Biologic Resource Center of Montpellier University Hospital (CRB-CHUM; http://www.chumontpellier.fr; Biobank ID: BB-0033-00031), and this study benefitted from the expertise of Dr. Benjamin Rivière (hepatogastroenterology sample collection) and Dr. Edouard Tuaillon (CRB-CHUM manager). Samples were obtained from livers of adult patients who underwent liver resections for medical reasons unrelated to our research program, or from anonymous donors when the liver was considered unsuitable for organ transplantation. The use of human specimens for scientific purposes was approved by the French National Ethics Committee. Written or oral informed consent was obtained from each patient or family prior to surgery. The clinical characteristics of the liver donors are presented in Appendix A. Hepatocytes were isolated by using a two-step perfusion protocol and cultured as described previously [25].

### 2.7. Real Time-Quantitative Polymerase Chain Reaction (RT-qPCR)

Total RNA was isolated and purified using “Zymo Research—Quick RNA mini prep” kit for LS174T-hPXR cells and using Trizol reagent (Thermo Fisher Scientific, Illkirch-Graffenstaden, France) for primary human hepatocytes and HepaRG according to the manufacturer’s protocol. The concentration and the purity of isolated RNA were measured using a spectrophotometer NanoVue (GE health care life sciences, Velizy-Villacoublay, France). Reverse transcription was performed with the qScript cDNA SuperMix (VWR, Fontenay-sous-Bois, France) for LS174-hPXR and HepaRG cells and using a random hexaprimer and the MMLV Reverse Transcriptase Kit (Life Technologies) for primary human hepatocytes. Quantitative polymerase chain reactions were performed using SYBR green (Qiagen, Les Ulis, France) and specific primers (Appendix A) with the LightCycler^®^-480 real-time PCR system (Roche Diagnostics, Meylan, France). The relative amount of RNA was calculated with the 2^ΔΔCT^ method, and gene expression was normalized using GAPDH or RPLP0. In Figure 5AB, the level of expression was compared with the mean level of the corresponding gene expression in DMSO-treated cells and expressed as *n*-fold ratio.

## 3. Results

### 3.1. Transactivation of PXR by Dabrafenib in Human Cancer Cell Lines

We first evaluated the potential transactivation of PXR by dabrafenib in several reporter cell lines. Since PXR ligands are usually characterized by marked cross-species differences, the ability of dabrafenib to activate several PXR orthologs was assessed in HG5LN control cells and in HG5LN cells stably overexpressing human, mouse or zebrafish GAL4(DBD)-PXR(LBD). Dabrafenib (3 μM) was compared to known PXR agonists such as SR12813 (3 μM), PCN (3 μM) or clotrimazol (1 μM). The results showed that dabrafenib was a strong and specific agonist of human PXR with a 5-fold increase in luciferase expression (Figure 1B). Conversely, dabrafenib could not or slightly activate the zebrafish or the mouse PXR LBD (Figure 1C,D). Specific agonist effects were also evidenced for SR12813 on the human PXR LBD and for PCN on the mouse PXR LBD, while clotrimazol strongly activated the zebrafish PXR (5-fold) and had only a moderate (2-fold) effect on human PXR LBD (Figure 1). The ability of dabrafenib to modulate the activity of hCAR was also assessed in HG5LN cells stably expressing human CAR WT or the mutant CAR (+APYLT) that displays a reduced basal activity. No agonist or antagonist activity of dabrafenib on hCAR was observed (Figure 1E,F).

We then performed dose–response experiments in different hPXR reporter cell lines to compare the agonist activity of dabrafenib to the activity of SR12813, and EC_50_ values were calculated (Figure 2). In HG5LN GAL4-hPXR cells, the EC_50_ value for dabrafenib (82 nM) was approximately 2-fold lower than that of SR12813 (239 nM), confirming the strong agonist potency of dabrafenib for hPXR (Figure 2A and Appendix A). Similar results were obtained in the human colon LS174T (EC_50_ of 110 nM), liver HepG2 (EC_50_ of 97.6 nM) and prostate 22RV1 (EC_50_ of 98.4 nM) carcinoma cell lines stably overexpressing hPXR (Figure 2B–D and Appendix A). SR12813-induced hPXR transactivation was also slightly different for all tested cell lines with EC_50_ values ranging from 125 to 434 nM (Appendix A). Similar results were obtained in terms of efficacy as dabrafenib behaved as a full agonist in all reporter cell lines. We observed a decrease of transactivation at concentrations higher than 0.3 μm that may be attributed to its toxicity, which was more pronounced in HepG2 cells than in the other cell lines (Figure 2). No induction of luciferase expression was observed in HG5LN control cells (Appendix A). Furthermore, the PXR antagonist SPA70 [25] could abrogate dabrafenib-induced luciferase expression in HG5LN GAL4-hPXR cells (Appendix A). Together, these results demonstrate that dabrafenib is a strong activator of the transcriptional activity of hPXR. One should also note that the hPXR antagonist SPA70 decreased the basal luciferase expression in HG5LN GAL4-hPXR cells (Appendix A), indicating that hPXR is partially activated in its apo form.

We next assessed the binding characteristics of dabrafenib to PXR using competitive binding assays with time-resolved fluorescence resonance energy transfer between a fluorescent PXR ligand and purified hPXR-LBD (LanthaScreen TR-FRET PXR Competitive Binding Assay). The results confirmed that dabrafenib directly binds to hPXR with an affinity slightly lower than that of SR12813 (Figure 3). The fact that in transactivation experiments dabrafenib is slightly more potent than SR12813 could indicate a potentially better bioavailability compared to SR12813.

Since PXR activation is known to stimulate proliferation of human colon cancer cell lines [16], we investigated whether dabrafenib could affect the growth of LS174T-hPXR cells. To that purpose, proliferations of LS174T control cells and LS174T-hPXR cells were compared in the absence or in the presence of the hPXR antagonist SPA70, SR12813 or dabrafenib using MTT assays (Figure 4A). As shown, in LS174T control cells, the proliferation was not affected by hPXR ligands. On the contrary, the basal proliferation of LS174T-hPXR cells was significantly reduced by the hPXR antagonist SPA70 indicating that hPXR is partially activated on proliferation in its apo form. Proliferation was also significantly increased by SR12813 and dabrafenib. Curiously, the efficacy of dabrafenib was higher than that of SR1813 (2.2-fold vs. 1.5-fold). Both activation of proliferation was abrogated in the presence of SPA70 confirming that this proliferation is mediated through hPXR.

We developed a new test that we referred to as the *P*-screen test to specifically assess compounds with hPXR activity that could induce cell proliferation (see Materials and Methods). To reduce the basal proliferation of the cells and allowing a more sensitive detection of any hPXR agonist activity that would result in growth induction, we pre-treated the cells with SPA70 100 nM. Then, the medium was removed, and cells were treated with increasing concentrations of dabrafenib and SR12813 (1 nM–3 μM) for 7 days. We confirmed that dabrafenib was more efficient than SR12813 at inhibiting the proliferation of LS174T-hPXR cells (4.56-fold vs. 3.2-fold). Conversely, the potency of both hPXR agonists was very similar (24 nM and 38 nM for dabrafenib and SR12813, respectively).

### 3.2. Effects of Dabrafenib on the Expression of hPXR Target Genes

The effects of dabrafenib and SR12813 on the expression of representative hPXR target genes involved in drug metabolism (CYP3A4, CYP2B6), drug transport (ABCG2) and cell proliferation (FGF19) were then measured by RT-qPCR in LS174T-hPXR and human normal hepatocytes (Figure 5). The results showed that the expression of hPXR target genes was increased in a concentration-dependent manner but was differentially modulated depending on the agonist. In LS174T-hPXR, CYP3A4 expression was increased by 190-fold with SR12813 as compared to 90-fold with dabrafenib. However, dabrafenib exhibited a higher potency as maximal induction of CYP3A4 expression was observed at 0.3 µM as compared to 1 µM for SR12813 (Figure 5A). The increase in ABCG2 expression was also higher when cells were treated with SR12813 as compared to dabrafenib (12-fold vs 9-fold, respectively, Figure 5A). Conversely, expression of FGF19 was drastically induced by dabrafenib (33-fold induction at 1 µM), whereas it was only increased by 4-fold with SR12813, even at the highest concentration used (3 µM). This higher efficacy of dabrafenib to induce FGF19 expression could explain the better efficacy of dabrafenib to activate the LS174T-hPXR proliferation. This is in agreement with the results of Wang et al. [16] suggesting FGF19 as the main mediator of PXR in colon cancer cell proliferation. Finally, using more biologically relevant primary cultures of freshly isolated human hepatocytes from three independent donors, we confirmed that both SR12813 and dabrafenib could significantly induce the expression of CYP3A4 and CYP2B6 genes and showed a similar potency (Figure 5B).

## 4. Discussion

A recommended dose for dabrafenib is 150 mg twice daily administered orally giving a plasma concentration of 2160 ng/mL [18]. After its administration, dabrafenib is predominantly metabolized by oxidation leading to hydroxy-dabrafenib and subsequently to carboxy-dabrafenib that undergoes a decarboxylation to generate desmethyl-dabrafenib, an active derivative that is further transformed to minor oxidative products [26]. CYP3A4 and CYP2C8 are known to play major roles in these metabolic steps, and dabrafenib was shown to induce CYP3A4 and CYP2D6 expressions in vitro and in vivo [20]. These studies have suggested that master regulators of genes involved in drug metabolism, such as the nuclear receptors hPXR or hCAR, could mediate the cell response to dabrafenib and influence the effects of other associated medications. However, there was no study characterizing the hPXR agonist activity of dabrafenib. The results of our study demonstrate for the first time that, similar to the reference hPXR agonist SR12813, dabrafenib is a selective activator of the hPXR in various cell lines overexpressing this nuclear receptor. For both dabrafenib and SR12813, discrepancies in hPXR transactivation could be noticed with regards to cell type. This could be attributable to a difference in the metabolic capacities of HepG2, LS174T, 22RV-1 and HeLa cells. We further demonstrated that dabrafenib-mediated hPXR activation was associated with enhanced expression of several hPXR target genes, which is in accordance with increased expression of CYP2B6 and CYP3A4 previously observed in hepatocytes [20]. Together, these results confirm that dabrafenib could regulate the expression of key enzymes involved in the metabolism of xenobiotics and demonstrate that this effect is linked to the hPXR agonist property of dabrafenib. These results are of clinical importance as dabrafenib could directly impact its own metabolism or the metabolism of other drugs or medications, in particular MEK inhibitors such as trametinib that is approved in combination with dabrafenib for the treatment of metastatic melanoma. In this line, we are currently investigating whether dabrafenib could increase the activity of metabolic enzymes in melanoma cells overexpressing hPXR and whether this could affect the efficacy of the co-treatment with trametinib.

While dabrafenib usually inhibits the proliferation of cancer cells by targeting BRAF^V600E^ [26], it was also shown to induce the proliferation of tumor cell lines expressing wild-type BRAF and mutant RAS [19]. In vitro studies reported half-inhibition concentration (IC_50_) values of 0.65 nM for BRAF^V600E^ ([18] and Appendix A). Here, we showed that dabrafenib could stimulate the growth of LS174T-hPXR cells with an EC_50_ of 28 nM. This effect was only observed when hPXR was stably overexpressed, strongly suggesting that this effect was, at least in part, mediated by hPXR and could also be attributable to an increase in FGF19 expression, a hPXR target gene known to be involved in cell proliferation [27] and that is significantly increased in LS174T-hPXR cells treated with dabrafenib. Interestingly, the PXR ligand rifampicin was also shown to induce FGF19-mediated proliferation of PXR-transfected LS174T cells [16]. At its plasma concentration (2160 ng/mL or 4.15 μM), dabrafenib is expected to be active on both BRAF^V600E^ and hPXR, indicating that the use of lower concentrations of dabrafenib could improve its selectivity while maintaining its efficacy. Another alternative would be to design dabrafenib analogues that would have the same affinity for BRAF but would not activate hPXR. We undertook this type of work and were able to synthesize dabrafenib analogs lacking PXR activity (Schneider et al., in preparation).

Since numerous pharmaceuticals like antiandrogens, antiestrogens, anti-epileptic or anti-viral drugs are ligands of hPXR [4,5,14], our results further emphasize the need to assess the potential agonist activity of drugs that are developed in a more thorough way. Indeed, former preclinical studies evaluating this activity were usually performed in mouse or rat models, but our results clearly demonstrate that dabrafenib did not show any significant activity towards the mouse ortholog of PXR. In order to improve the assessment of PXR agonist activity, we therefore recommend to use adapted cell models to screen for hPXR agonist activity, and we propose a panel of in vitro techniques enabling such an evaluation for new developed drugs. Those original models could also serve to design new analogs of currently used drugs devoid of hPXR binding and activation.

## Figures and Tables

**Figure 1 cells-09-01641-f001:**
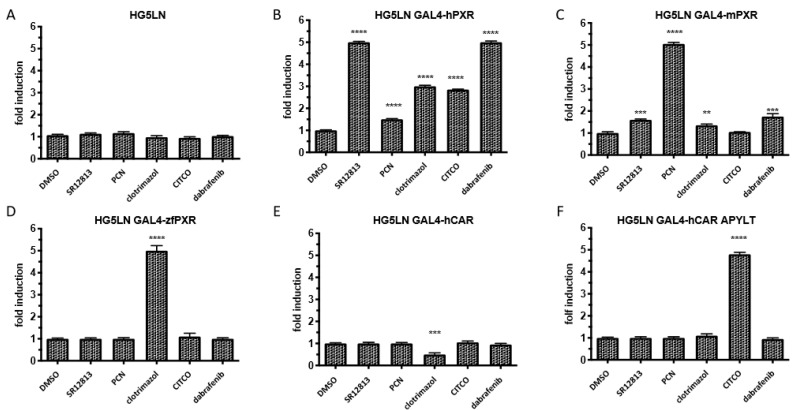
Effects of dabrafenib on the activation of human, mouse and zebrafish pregnane X receptor (PXR) and human constitutive androstane receptor (CAR) nuclear receptors as measured by luciferase reporter assay. Activation has been measured in (**A**) HG5LN control cells or cells expressing GAL4 fusion with (**B**) human (hPXR), (**C**) mouse (mPXR) or (**D**) zebrafish (zfPXR) PXR, (**E**) human wild-type (hCAR) or (**F**) mutated (hCAR APYLT) hCAR LBD treated by DMSO (0.1%), SR12813 3 μM, PCN 3 μM, clotrimazol 1 μM, CITCO 1 μM and dabrafenib 3 μM. Results are expressed as fold induction as compared to control. Data are expressed as the mean ± sd of three independent experiments, **** *p* < 0.0001, *** *p* < 0.001, ** *p* < 0.01 (Student’s *t*-test) compared with DMSO-treated cells.

**Figure 2 cells-09-01641-f002:**
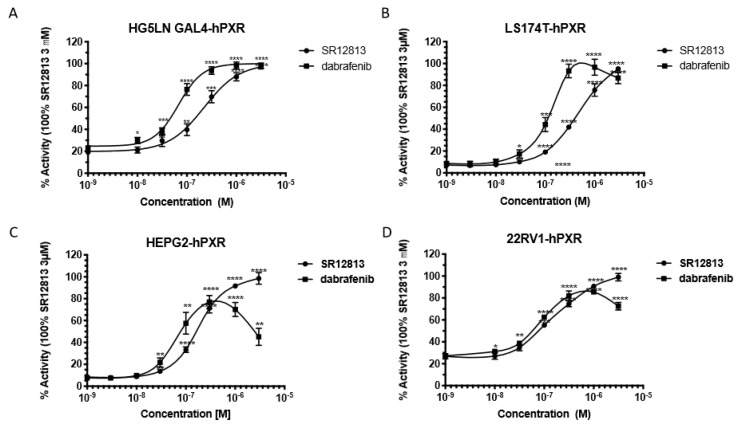
Dabrafenib is a potent activator of hPXR on different human cancer cell lines. Results of luciferase assays showing dose–response curves for SR12813 and dabrafenib in (**A**) HG5LN GAL4-hPXR, (**B**) LS174T-hPXR, (**C**) HepG2-hPXR and (**D**) 22RV1-hPXR reporter cell lines. Results are expressed as a percentage of the maximal response obtained in the presence of 3 μM SR12813. Data are the mean ± sd of three to five independent experiments, **** *p* < 0.0001, *** *p* < 0.001, ** *p* < 0.01, * *p* < 0.1 (Student’s *t*-test) compared with DMSO-treated cells.3.2. Time-Resolved Fluorescence Energy Transfer Experiments.

**Figure 3 cells-09-01641-f003:**
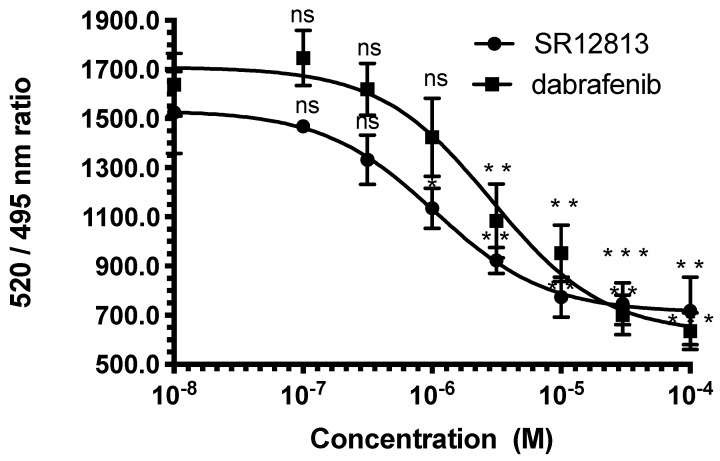
Dabrafenib binds to hPXR. Inhibition of FRET between fluorescein-labeled PXR ligand and recombinant GST-hPXR by SR12813 or dabrafenib. Results are expressed as the signal from the fluorescein emission divided by the terbium signal to provide a TR-FRET emission ratio. Data are the mean (±sd) from three independent experiments, *** *p* < 0.001, ** *p* < 0.01, * *p* < 0.1 (Student’s *t*-test) compared with DMSO-treated cells.3.3. Induction of hPXR-Mediated Proliferation by Dabrafenib.

**Figure 4 cells-09-01641-f004:**
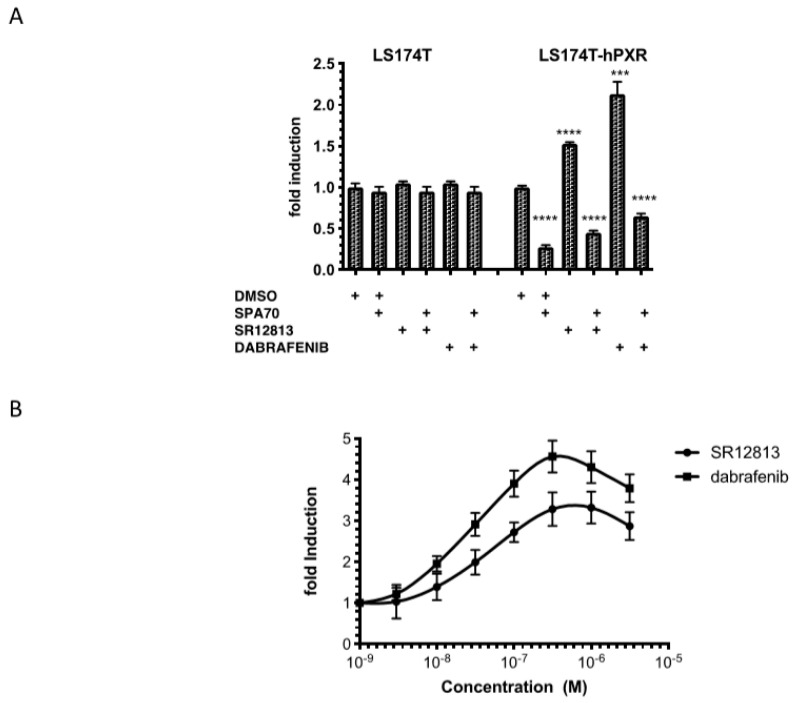
Effects of dabrafenib and SR12813 on the proliferation of LS174T-hPXR cells. (**A**). LS174T control cells and LS174T hPXR cells were treated with 0.1% DMSO, 3 μM SPA70, 0.3 μM SR12813, 0.3 μM SR12813 + 3 μM SPA70, 0.3 μM dabrafenib and 0.3 μM dabrafenib + 3 μM SPA70 for 7 days. (**B**) LS174T hPXR cells were pretreated 24 h with SPA70 100 nM. This pretreatment was performed in order to reduce the basal proliferation of the cells, thus allowing a more sensitive detection of any PXR agonist activity that would result in growth induction. Then, medium was removed, cells were treated with increasing concentrations of SR12813 or dabrafenib for 7 days continuously, and cell growth was measured using MTT assay. Data are expressed as fold change in cell growth as compared to untreated cells and expressed as mean ± sd of three to six independent experiments, **** *p* < 0.0001, *** *p* < 0.001 (Student’s *t*-test) compared to DMSO-treated cells.

**Figure 5 cells-09-01641-f005:**
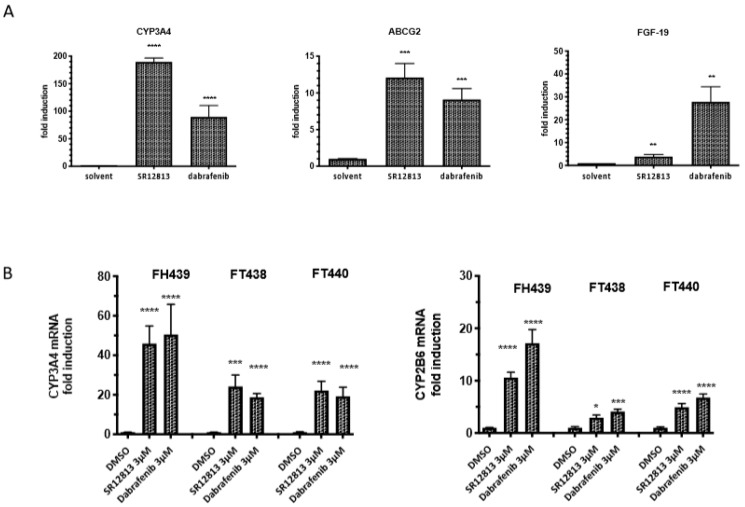
Effects of dabrafenib on hPXR target genes expression in LS174T-hPXR cells and in primary human hepatocytes. (**A**) RT-qPCR of CYP34A, ABCG2 and FGF19 mRNA expression were performed in LS1747-hPXR cells treated with 3 μM SR12813 or dabrafenib for 24 h. Results were obtained from three independent experiments performed in duplicate. Data are expressed as mean ± sd and compared with control cells treated with DMSO (0.1%). (**B**) RT-qPCR of CYP34A and CYP2B6 mRNA expression in primary cultures of human hepatocytes (three independent donors: FT438, FH439 and FH440) following 48 h treatment with the indicated concentrations of ligand. The relative gene expression levels were normalized using GAPDH content for LS174T-hPXR cells and for RPLP0 for human hepatocytes. mRNA expressions are expressed as mean ± sd of three independent experiments and compared with DMSO-treated LS174T-hPXR cells or hepatocytes, **** *p* < 0.0001, *** *p* < 0.001, ** *p* < 0.01, * *p* < 0.1 (Student’s *t*-test).

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
