# Peer review of "The Anti-Cancer Drug Dabrafenib Is a Potent Activator of the Human Pregnane X Receptor"

_cells, 2020, doi:10.3390/cells9071641_

Round 1

Reviewer 1 Report

Summary of manuscript:

In this manuscript, the effects of cancer drug dabrafenib were studied on receptors CAR and PXR and some other cellular functions. These receptors control the expression of drug metabolizing enzymes, which could cause drug-drug interactions. Receptor activation, cell proliferation, and mRNA expression were all measured. There were a detailed number of experiments conducted. It is not clear how the concentrations used in this study are related to in vivo concentrations, which is a very important fact to make the study more relevant. There were many errors in the figures and text, some of which are detailed below. The proliferation experiment does not really match the theme of the manuscript, and its relevance is also weak because PXR was overexpressed in the cells. The authors must proofread their entire manuscript to make sure it is accurate. There were missing tables as well. The comments below may help the authors improve their manuscript.

Comments:

  1. Table 1, Supplementary Table 1, and Supplementary Table 2 were mentioned in the manuscript text but these are all missing.

  1. The prescribing information for dabrafenib (found at https://www.novartis.us/sites/www.novartis.us/files/tafinlar.pdf) says the following: “In vitro data demonstrate that dabrafenib is an inducer of CYP3A4 and CYP2B6 via activation of the pregnane X receptor (PXR) and constitutive androstane receptor (CAR) nuclear receptors.” (a) Therefore, please explain in the manuscript how this study provides new information. (b) Also please explain in the manuscript how your data contradicts with existing data regarding CAR activation.

  1. In the abstract the authors should make background information more concise, and add more data and results. The abstract should be comprehensive and cover all major results like gene expression, primary hepatocytes, and different species of PXR. There should be a logical order of experiments and their results. Quantiative data must be included to support the statements.

  1. Figure 4 shows cell proliferation increasing due to exposure to dabrafenib and SR12813. However, the cells contain high amount of overexpressed PXR, an artificial situation. This makes the purpose and relevance of the experiment weak, especially given the statement in the discussion that links this data with colon cancer (lines 301-302). Another major drawback of this study is that it does not align with the theme of the paper, which focuses on receptor activation, gene expression, drug metabolizing enzymes. Therefore, for both of these reasons this data should be eliminated from the manuscript. This will yield a more clear theme and avoid experiments that yield irrelevant data.

  1. (a) Please label all individual graphs with separate letters (and update the captions) so the figures are consistent and easier to understand. (b) Also, in all figures be consistent when refering to negative controls (call them DMSO or solvent but don’t use both terms).

  1. In the discussion, please explain how these results, at the concentrations indicated, are related to plasma concentrations or other in vivo concentrations of the drug. This info is very important to make the study more relevant.

  1. Section 3.1 says PXR from different species were expressed in HeLa cells but Figure 1 caption says HG5LN cells were used. Please correct this contradiction throughout the manuscript.

  1. For Figure 2, where is data showing control cells that do not contain transfected receptors.

  1. Initial experiments of dabrafenib binding to PXR and CAR used 1 uM concentration. In the manuscript please justify why this concentration was selected and how it relates to in vivo concentrations.

  1. In section 2.1, please include the full name of CITCO and PCN.

  1. In the methods section please provide the equations for dose-response curves because there are different variations of the equation (e.g., fixed vs. variable values of top, bottom, slope/Hill coefficient).

  1. Introduction, last paragraph (lines 70-72): please modify this text to provide an overview of the experiments conducted instead of the results. For example, “in this study, we measured…”

  1. Although receptor activation and gene expression were measured, neither of these directly relate to drug-drug interactions because these depend on the activity of the metabolic enzymes, which was not measured. Please add this explanation to the discussion.

  1. Lines 189-191: Please mention this data is in Supplementary Figure 1.

  1. Lines 203-204: Please eliminate this sentence that talks about bioavailability because this study has no connection to bioavailability.

  1. Figure 1: (a) please change caption text from “zPXR” to “zfPXR” to be consistent. (b) The y-axes should be labeled as “fold induction” not “percent induction”. (c) Two of the graphs are labeled as “hCAR APYLT” (one of these should be “hCAR”). (d) In the first sentence of the caption, provide a better summary by saying this figure shows species differences of PXR and CAR activation.

  1. Figure 2: (a) please change the x-axis from log values (-5, -6, -7, etc…) to a log scale (10^-5, 10^-6, 10^-7, etc…) so all graphs are consistent. (b) Also, please label each graph (A, B, C, D). (c) In the first sentence of the caption, provide a better summary by saying this figure shows cell line differences of human PXR activation.

  1. Figure 4A and 4B: (a) please label y-axis as “fold” induction or proliferation (not percent). (b) Some text on x-axis is not legible, please simplify.

  1. Figure 5B: (a) Please combine CYP3A4 graphs and show the same concentrations among the three donors. These graphs were hard to understand because they have different concentrations and were not consistent. (b) For CYP2B6 graph, please include data for donor FT438 because it was missing.

  1. Supplementary Figure 1: the y-axis should be “fold” not percent.

  1. Page 8, lines 276-277: Please eliminate “However, there was no evidence..” because it is already known that dabrafenib binds to PXR (please see earlier comment).

  1. Line 281: Please remove the term “pharmacokinetics”, because this is what happens in vivo (adsorption, metabolism, distribution, excretion). The differences observed among cell lines may be due to metabolism only, which is only one aspect of pharmacokinetics.

  1. Line 289: please replace “infratoxic” with another word. This word does not have a definition when searching online.

Author Response

Reviewer 1

In this manuscript, the effects of cancer drug dabrafenib were studied on receptors CAR and PXR and some other cellular functions. These receptors control the expression of drug metabolizing enzymes, which could cause drug-drug interactions. Receptor activation, cell proliferation, and mRNA expression were all measured. There were a detailed number of experiments conducted.

It is not clear how the concentrations used in this study are related to in vivo concentrations, which is a very important fact to make the study more relevant. There were many errors in the figures and text, some of which are detailed below.

The proliferation experiment does not really match the theme of the manuscript, and its relevance is also weak because PXR was overexpressed in the cells.

The authors must proofread their entire manuscript to make sure it is accurate. There were missing tables as well. The comments below may help the authors improve their manuscript.

First of all, we would like to thank the Reviewer for her/his thorough evaluation of our data and her/his valuable and constructive suggestions that allowed us to improve the manuscript. We also apologize for the errors that were made. We have corrected them in the revised version of the manuscript and proofread the entire manuscript to ensure the accuracy of the data.

Regarding dabrafenib concentrations used in the study, we would like to point out that dabrafenib is a drug with two distinct effects: one is linked to its anticancer activity and is related to the inhibition of the serine-threonine kinase BRAFV600E with an in vitro IC50 of 0.65 nM as shown in the new Supplementary Figure 2 of the revised manuscript and in the review by Puszkiel et al, 2018; the other effect is unwanted and is linked to hPXR activation with an in vitro EC50 in the 100 nM range (Figure 2 and Supplementary Table 4 of the revised manuscript). Dabrafenib (Tafinlar, GSK2118436, MW 519.56) is prescribed at a recommended dose of 150 mg twice daily administered orally, giving a plasma concentration of 2,160 ng/ml (4 mM) (Puszkiel et al, 2018). At this concentration, dabrafenib is expected to be active on both BRAFV600E and PXR. Therefore, anticancer efficacy of dabrafenib cannot be dissociated from its agonistic activity towards PXR, unless it is administered at reduced doses. We have clarified this concept in the revised version of the manuscript.

With respect to the proliferation test, we have proposed it as a complementary assay to further demonstrate that dabrafenib can bind and activate PXR. We agree with the Reviewer that PXR is overexpressed in LS174T colon cancer cells and that the level of expression is high. Based on qRT-PCR experiments, it is 5 to 10 times higher in LS174T-PXR compared to than that of human hepatocytes or differentiated HepaRG cells (Supplementary Figure 3 of the revised manuscript). Though PXR is overexpressed in this model, our results clearly demonstrate that dabrafenib-induced proliferation is PXR-specific as this effect is limited in LS174T cells with low (or no) expression of PXR. This test that we called P-test (in reference of the estrogenic MCF-7 proliferation called E-test) can therefore be used to determine whether a compound is a PXR agonist ligand or not. For these reasons, we think that these data should be included in the manuscript.

  1. Table 1, Supplementary Table 1, and Supplementary Table 2 were mentioned in the manuscript text but these are all missing.

We apologize for these omissions. Supplementary Tables 1 and 2 are now added in the revised Supplementary Materials.

  1. The prescribing information for dabrafenib (found at https://www.novartis.us/sites/www.novartis.us/files/tafinlar.pdf) says the following: “In vitro data demonstrate that dabrafenib is an inducer of CYP3A4 and CYP2B6 via activation of the pregnane X receptor (PXR) and constitutive androstane receptor (CAR) nuclear receptors.” (a) Therefore, please explain in the manuscript how this study provides new information. (b) Also please explain in the manuscript how your data contradicts with existing data regarding CAR activation.

  1. a) As mentioned by the Reviewer, we were aware of the prescribing information for dabrafenib that was accessible on the Novartis website. However, we could not find any scientific publication providing experimental evidences supporting the Novartis note. This is indeed one of the main reasons for which we performed our study. Besides the fact that robust experimental data should be available for the scientific community, we also provide novel data regarding the characterization of the PXR activity of dabrafenib (potency, efficacy and affinity) that is consistent with the scope of the Cells special edition dedicated to PXR.

  1. b) We also tested dabrafenib for its ability to activate HG5LN GAL4-hCAR (Fig 1) and GAL4-mCAR (data not shown) and we did not observed any modulation of their activities, which is in disagreement with the prescribing information from Novartis though no data have been disclosed by the company to support CAR activation. While we cannot explain this apparent contradiction our experimental evidences strongly suggest that dabrafenib could be an inducer of CYP3A4 and CYP2B6 mainly through PXR.

.

  1. In the abstract the authors should make background information more concise, and add more data and results. The abstract should be comprehensive and cover all major results like gene expression, primary hepatocytes, and different species of PXR. There should be a logical order of experiments and their results. Quantitative data must be included to support the statements.

We modified the abstract as suggested by the Reviewer remark, rendering it more concise and covering the major results to strengthen our statements.

  1. Figure 4 shows cell proliferation increasing due to exposure to dabrafenib and SR12813. However, the cells contain high amount of overexpressed PXR, an artificial situation. This makes the purpose and relevance of the experiment weak, especially given the statement in the discussion that links this data with colon cancer (lines 301-302). Another major drawback of this study is that it does not align with the theme of the paper, which focuses on receptor activation, gene expression, drug metabolizing enzymes. Therefore, for both of these reasons this data should be eliminated from the manuscript. This will yield a more clear theme and avoid experiments that yield irrelevant data.

We are sorry but we did not completely agree with the reviewer on these different points. While we agree with the Reviewer upon the fact that we artificially overexpressed PXR in LS174T-PXR colon cancer cells, However, the positive effect of PXR ligands (rifampicin) effect was also observed in LS174T cells without overexpression of PXR by Wang et al (J Clin Invest 2011). In our LS174T native cells and probably due to a too low expression of PXR (supplementary figure 3 in the revised paper), the effect on proliferation of PXR agonists is negligible (Figure 4 of the paper). To increase this effect, we needed to overexpress PXR.

As mentionned above, we measured the level of PXR expression by RT-PCR experiments in the different cells used in our study, LS174T, LS174T-PXR, hepatocytes and differentiated HepaRG cells. The results are illustrated in the supplementary fig 3 of the revised article.

According to the reviewer and because the effect on proliferation was only observed in cells overexpressing PXR, we removed the sentence suggesting that dabrafenib can increase the risk to develop a colon cancer in patients treated for a melanoma.

  1. (a) Please label all individual graphs with separate letters (and update the captions) so the figures are consistent and easier to understand. (b) Also, in all figures be consistent when refering to negative controls (call them DMSO or solvent but don’t use both terms).

According to the Reviewer’s suggestions, we a) modified the graphs and the figures and b) used only DMSO with reference to negative controls.

  1. In the discussion, please explain how these results, at the concentrations indicated, are related to plasma concentrations or other in vivo concentrations of the drug. This info is very important to make the study more relevant.

As mentioned in our response to the first point that was raised by the Reviewer, we explained how the results have been obtained with concentrations that are clinically relevant with plasma concentrations. This important information is now mentioned both in the Introduction and in the Discussion sections of the revised manuscript.

  1. Section 3.1 says PXR from different species were expressed in HeLa cells but Figure 1 caption says HG5LN cells were used. Please correct this contradiction throughout the manuscript.

We are sorry for this mistake and have corrected this contradiction in the figure and the text of the revised manuscript.

  1. For Figure 2, where is data showing control cells that do not contain transfected receptors.

The absence of effect of dabrafenib on HG5LN control cells was shown in new Supplementary Figure 1.

  1. Initial experiments of dabrafenib binding to PXR and CAR used 1 uM concentration. In the manuscript please justify why this concentration was selected and how it relates to in vivo concentrations.

We apologize for this mistake. Initial experiments used dabrafenib at 3 µM concentration. At higher concentrations, we observed that a 24h treatment with dabrafenib could inhibit luciferase expression.

  1. In section 2.1, please include the full name of CITCO and PCN.

As suggested, we included the full name of CITCO and PCN (line 140, manuscript_ revised_marked).

  1. In the methods section please provide the equations for dose-response curves because there are different variations of the equation (e.g., fixed vs. variable values of top, bottom, slope/Hill coefficient).

We added the following information in the revised manuscript (line 234).

The equation used by Graphpad was Y=Bottom + (Top-Bottom)/(1+10^((LogEC50-X)*HillSlope)).

X is the log of concentration. Y is the response.

As an example, for Figure 2, the bottom is the response in absence of ligand and the top (100%) is the response in presence of the reference agonist at saturating concentrations (SR12813 3 mM for hPXR).

  1. Introduction, last paragraph (lines 70-72): please modify this text to provide an overview of the experiments conducted instead of the results. For example, “in this study, we measured…”

As suggested by the Reviewer, we modified the end of the introduction in the revised version of the manuscript in order to provide an overview of the experiments instead of the results (line 92).

  1. Although receptor activation and gene expression were measured, neither of these directly relate to drug-drug interactions because these depend on the activity of the metabolic enzymes, which was not measured. Please add this explanation to the discussion.

We agree with the Reviewer comment and introduced this explanation at the end of the first paragraph of the Discussion (line 483).

  1. Lines 189-191: Please mention this data is in Supplementary Figure 1.

As suggested, we mentioned that this data is shown in Supplementary Figure 1

  1. Lines 303-304: Please eliminate this sentence that talks about bioavailability because this study has no connection to bioavailability.

As suggested, the sentence was deleted (line 474).

  1. Figure 1: (a) please change caption text from “zPXR” to “zfPXR” to be consistent. (b) The y-axes should be labeled as “fold induction” not “percent induction”. (c) Two of the graphs are labeled as “hCAR APYLT” (one of these should be “hCAR”). (d) In the first sentence of the caption, provide a better summary by saying this figure shows species differences of PXR and CAR activation.

These changes were performed in accordance with the Reviewer recommendations.

  1. Figure 2: (a) please change the x-axis from log values (-5, -6, -7, etc…) to a log scale (10^-5, 10^-6, 10^-7, etc…) so all graphs are consistent. (b) Also, please label each graph (A, B, C, D). (c) In the first sentence of the caption, provide a better summary by saying this figure shows cell line differences of human PXR activation.

These changes were performed in accordance with the Reviewer recommendations.

  1. Figure 4A and 4B: (a) please label y-axis as “fold” induction or proliferation (not percent). (b) Some text on x-axis is not legible, please simplify.

The y-axis was labeled as fold-induction and we simplified the text on the x-axis.

  1. Figure 5B: (a) Please combine CYP3A4 graphs and show the same concentrations among the three donors. These graphs were hard to understand because they have different concentrations and were not consistent. (b) For CYP2B6 graph, please include data for donor FT438 because it was missing.

The changes were made according to the Reviewer’s recommendations..

  1. Supplementary Figure 1: the y-axis should be “fold” not percent.

The y-axis labeling of Supplementary Figure 1 was changed to « fold ».

  1. Page 8, lines 276-277: Please eliminate “However, there was no evidence..” because it is already known that dabrafenib binds to PXR (please see earlier comment).

We replaced the sentence ‘’However, there was no evidence demonstrating the potential PXR agonist activity of dabrafenib ‘’ by ‘’However, there was no published study characterizing the PXR agonist activity of dabrafenib.’’ (line 470).

  1. Line 281: Please remove the term “pharmacokinetics”, because this is what happens in vivo (adsorption, metabolism, distribution, excretion). The differences observed among cell lines may be due to metabolism only, which is only one aspect of pharmacokinetics.

According to the Reviewer’s comments, we have removed the term pharmacokinetics and modified the sentences as follows: For both dabrafenib and SR12813, discrepancies in hPXR transactivation could be noticed with regards to cell type. This could be attributable to a difference in metabolic capacity of HepG2, LS174T, 22RV-1 and HeLa cells. (line 474)

.

  1. Line 289: please replace “infratoxic” with another word. This word does not have a definition when searching online.

This term has been removed (line 489).

Reviewer 2 Report

The manuscript describes a study investigating the capacity of an anticancer drug dabrafenib to directly activate the human pregnane X receptor (hPXR) and the constitutive androstane receptor (CAR) .

Dabrafenid has been described as an inducer of various CYP activities may be through hPXR or h CAR mediated transcriptional regulation.  But no study has been performed to evaluate their capacity to directly activate these nuclear receptors.

 As described by the authors these nuclear receptors play a critical role in the control of detoxifying enzymes involved in the metabolism and clearance of drugs in the liver and gastrointestinal tract. They could be thus involved in the drug-drug interactions. As stated by authors it is thus critical to take into account the potential of hPXR and hCAR activation in the rational development of anticancer drug combinations to limit DDIs and reduce the risk of adverse events.

This is a very original study well conducted and well-controlled. The manuscript is well written. This manuscript is publishing in Cells but some revisions and clarifications are needed

  • Introduction authors should add the recent paper of Puszkiel A. 2018 (clinical pharmacokinetics) that describes the clinical pharmacokinetics and pharmacodynamics of dabrafenib

  • Material and methods: authors must better state the experimental design. It is not clear whether the transactivation , the proliferation assays realized on same or different cell passages and whether this could induce some bias
  • Results
    1. in figure 2 and 3 authors have to explain why there is no statistical estimation of the observed variations
    2. What is the biological relevance of the assessed Dabrafenib dose compared to human treatment?
    3. authors must better described the P-screen test design and make an effort to better explain the results (from line 221 to line 229) because it is difficult for the reader to understand the results in this paragraph and its reliability from the figures presented
    4. Authors must explain the reason for the choice of the assessed cancerous cell lines (colon and prostate). Authors should also explain why they don’t assess the impact of drafenib on cancerous cells from lung and/or  thyroid since this drug is dedicated for this type of cancer
    5. line 188 when authors state “ “Together, these results demonstrate that dabrafenib is a strong activator of the transcriptional activity of human PXR in human cancer cell models” their results cannot be generalized to all the cancerous cells since their models are  reporter cell lines and authors must moderate this conclusion
    6. the paper could be improved by evaluating the interaction between dabrafenib and trametinib on hPXR activation, trametinib being the drug commonly used in combination with dabrafenid in cancerous patients . It is all the more appropriate given that in the introduction authors stated that PXR and CAR could be responsible for DDIs when drugs are co-administered.
    7. What about the impact of a chronical exposure to dabrafenid on the activation of h PXR?
    8. Authors must precise whether donors were male or female since there are often sexual dimorphic response regarding to detoxifying pathway.

Author Response

The manuscript describes a study investigating the capacity of an anticancer drug dabrafenib to directly activate the human pregnane X receptor (hPXR) and the constitutive androstane receptor (CAR).

Dabrafenib has been described as an inducer of various CYP activities may be through hPXR or h CAR mediated transcriptional regulation.  But no study has been performed to evaluate their capacity to directly activate these nuclear receptors.

 As described by the authors these nuclear receptors play a critical role in the control of detoxifying enzymes involved in the metabolism and clearance of drugs in the liver and gastrointestinal tract. They could be thus involved in the drug-drug interactions. As stated by authors it is thus critical to take into account the potential of hPXR and hCAR activation in the rational development of anticancer drug combinations to limit DDIs and reduce the risk of adverse events.

This is a very original study well conducted and well-controlled. The manuscript is well written. This manuscript is publishing in Cells but some revisions and clarifications are needed.

We would like to thank the Reviewer for her/his comments stipulating the originality of our study.

  1. Introduction authors should add the recent paper of Puszkiel A. 2018 (clinical pharmacokinetics) that describes the clinical pharmacokinetics and pharmacodynamics of dabrafenib.

We thank the Reviewer for this suggestion and added this pertinent review as a new reference in the revised manuscript (line 83, 463, 490).

  1. Material and methods: authors must better state the experimental design. It is not clear whether the transactivation, the proliferation assays realized on same or different cell passages and whether this could induce some bias.

As suggested by this Reviewer, information about the reporter cell lines and their stability and the reproducibility of the results using these models was added in the Material and Methods section of the revised manuscript (line 172 and 227).

  1. Results
    1. in figure 2 and 3 authors have to explain why there is no statistical estimation of the observed variations

We added statistical estimation to figures 2 and 3.

  1. What is the biological relevance of the assessed Dabrafenib dose compared to human treatment?

We have addressed this important point (that was also raised by Reviewer 1) by adding the existing information on dabrafenib plasma concentrations in treated patients. This concentration (2160 ng/ml, 4 µM, Puszkiel et al, 2018) is active on both BRAFV600E and PXR and is compatible with the concentrations of dabrafenib used in our study, demonstrating the clinical relevance of our data (line 462).

  1. authors must better described the P-screen test design and make an effort to better explain the results (from line 221 to line 229) because it is difficult for the reader to understand the results in this paragraph and its reliability from the figures presente.

As suggested by the Reviewer, we better explained the P-screen and the necessity to pretreat the LS-PXR cells by the PXR antagonist SPA70 in order to reduce the basal activity of PXR and increase the fold induction of proliferation by PXR agonists (line 392, 417).

  1. Authors must explain the reason for the choice of the assessed cancerous cell lines (colon and prostate). Authors should also explain why they don’t assess the impact of dabrafenib on cancerous cells from lung and/or thyroid since this drug is dedicated for this type of cancer.

The aim of our work was to provide robust experimental evidences that dabrafenib binds to and activates hPXR. To this end, we tested dabrafenib in the PXR reporter cell lines that we have established in our team and validated in previous studies (Lemaire et al, Mol Pharmacol 2007 ; Delfosse et al, Nat Commun 2015 ; Grimaldi et al, Toxicol 2019). These cells are listed in Supplementary Table 1. Though we agree with the Reviewer’s comment that lung, melanoma, or thyroid models may be pertinent, we have not established PXR reporter lines in these contexts so far.

  1. line 188 when authors state “ “Together, these results demonstrate that dabrafenib is a strong activator of the transcriptional activity of human PXR in human cancer cell models” their results cannot be generalized to all the cancerous cells since their models are  reporter cell lines and authors must moderate this conclusion.

We agree with the Reviewer’s remark and removed ‘’in human cancer cell models’’ from the sentence (line 353).

  1. the paper could be improved by evaluating the interaction between dabrafenib and trametinib on hPXR activation, trametinib being the drug commonly used in combination with dabrafenid in cancerous patients. It is all the more appropriate given that in the introduction authors stated that PXR and CAR could be responsible for DDIs when drugs are co-administered.

We agree with the reviewer that it is important to evaluate the interaction between dabrafenib and Trametinib on hPXR activation. In a first screening, we tested more than forty anticancerous chemicals and among them dabrafenib is the more potent.   Trametinib was not active on hPXR. In this article, we wanted to characterize the interaction of dabrafenib with hPXR.

Our future work will be to study the antiproliferative activity of dabrafenib alone, trametinib alone or dabrafenib combined with trametinib in melanoma cells (A375 cells) overexpressing or not hPXR. We are currently establishing A375-hPXR cells. This work will enable us to evaluate the interaction between dabrafenib and trametinib on hPXR activation and reciprocally evaluate the effect of the PXR activation on the antiproliferative efficacy of dabrafenib and trametinib.

  1. What about the impact of a chronical exposure to dabrafenid on the activation of h PXR?

While it is an interesting point, we have not directly assessed this aspect of the problem. However, we have treated the cells for different exposure times (24 hours, 48 hours and seven days) and have not found significant differences in luciferase induction in both absolute values and fold induction.

  1. Authors must precise whether donors were male or female since there are often sexual dimorphic response regarding to detoxifying pathway.

Information concerning the sex of the three donors is now added in Supplementary Table 2 in the revised version of the manuscript.

Reviewer 3 Report

Although the manuscript could be interesting for readers, two important aspects limit its publication. The first is that the cellular systems used should be described and schematized appropriately, in order to provide readers, even if non-experts in the field, easily understanding of the used methodology. The second aspect, more important than the first one, is that the authors should demonstrate that dabrafenib treatment induces the expression of the receptor and its activation. For example, they could carry out real time PCR experiments following the treatment or better western blots analysis in both LS174T colon cancer cell line, HepG2 liver cancer cell line and in normal hepatocytes.

Other concerns are:

  1. In the introduction section more evidences for hCAR should be provided.
  2. References 6-13 provide evidences also for prostate and ovarian cancer. Please add these information’s in the main text.
  3. In the introduction authors reported that Dabrafenib is actually used for metastatic melanoma. Unfortunatly, their experiments were performed on different cancer cell lines from differen tumors (cervical, liver, prostate and colon). Please clarify.
  4. A scheme reporting the cellular constructs used in the present study should be added.
  5. Check if the reference number 20 is appropriately inserted in the text.
  6. Supplementary table 1 and table 2 are missing in supplementary files.
  7. It is the reviewer's opinion that the experiments in figure 1 reporting the transactivation results on mouse and zebrafish PRX, do not provide any information useful for the experimental design and should be deleted. Is more interesting to evaluate by western blot the PXR overexpression
  8. Please check Y and X axes in figure 2. Activity instead than Activitè, [M] or (M),and 10-9 instead than -9, etc.
  9. References should be appropriately formatted.

Author Response

Although the manuscript could be interesting for readers, two important aspects limit its publication. The first is that the cellular systems used should be described and schematized appropriately, in order to provide readers, even if non-experts in the field, easily understanding of the used methodology. The second aspect, more important than the first one, is that the authors should demonstrate that dabrafenib treatment induces the expression of the receptor and its activation. For example, they could carry out real time PCR experiments following the treatment or better western blots analysis in both LS174T colon cancer cell line, HepG2 liver cancer cell line and in normal hepatocytes.

We would like to thank the Reviewer for her/his insightful comments. Concerning the first point, we agree upon the fact that reporter cell lines that we used to characterize the PXR activity of dabrafenib were not sufficiently described. These cell lines are now summarized in Supplementary Table 1 and we have also added our previous studies concerning their specific use as new references.

Regarding the second point, we apologize for not sufficiently explaining that the action of dabrafenib is to bind to and activate PXR. Actually, dabrafenib has no effect on the expression of PXR, but its binding to PXR leads to an activation of the nuclear receptor and an increase in the expression of PXR target genes, i.e. luciferase in the reporter cell lines, or CYP3A4, CYP2B6, ABCG2 and FGF-19 in LS174T-PXR cells and human hepatocytes.

Other concerns are:

  1. In the introduction section more evidences for hCAR should be provided.

As we found that CAR is not a target of dabrafenib, we only mention hCAR in the beginning of the introduction and then focused on hPXR which is for us the main target of dabrafenib.

  1. References 6-13 provide evidences also for prostate and ovarian cancer. Please add these information’s in the main text.

We added this information in the text in accordance with the Reviewer’s remark (line 70).

  1. In the introduction authors reported that Dabrafenib is actually used for metastatic melanoma. Unfortunatly, their experiments were performed on different cancer cell lines from different tumors (cervical, liver, prostate and colon). Please clarify.

As it was also mentioned to Reviewer 2, the aim of our work was to provide robust experimental evidences that dabrafenib binds to and activates hPXR. To this end, we tested dabrafenib in the PXR reporter cell lines that we have established in our team and validated in previous studies (Lemaire et al, Mol Pharmacol 2007; Delfosse et al, Nat Commun 2015; Grimaldi et al, Toxicol 2019). These cells are listed in Supplementary Table 1. At present we have not establish a hPXR melanoma reporter cell line. However, we use A375 melanoma cells to test the antiproliferative effect of dabrafenib (by inhibition of BRAFV600E (Supplementary Figure 2).

  1. A scheme reporting the cellular constructs used in the present study should be added.

We did not add this scheme in our revised manuscript, but we cited in the references our former review (Grimaldi et al, Reporter cell lines for the characterization between human nuclear receptors and endocrine disruptors, Frontiers in Endocrinology 2015) (line 105) in which such as general scheme was presented.

  1. Check if the reference number 20 is appropriately inserted in the text.

We removed this reference which was not appropriately inserted in the text.

  1. Supplementary table 1 and table 2 are missing in supplementary files.

These Tables are added in the revised version of the manuscript.

  1. It is the reviewer's opinion that the experiments in figure 1 reporting the transactivation results on mouse and zebrafish PXR, do not provide any information useful for the experimental design and should be deleted. Is more interesting to evaluate by western blot the PXR overexpression

Mouse and zebrafish are often used to test the in vivo activity of pharmaceuticals. In figure 1, we wanted to demonstrate that dabrafenib only binds and activates human PXR and had no effect on mouse and zebrafish PXR. Because the PXR ligand binding domain is very different between species, these data show that classical in vivo models may not be adapted to the screening of the agonist activity of PXR, explaining why dabrafenib (and other pharmaceuticals) were not identified as hPXR ligands.

  1. Please check Y and X axes in figure 2. Activity instead than Activitè, [M] or (M),and 10-9instead than -9

These changes were made as suggested by the Reviewer.

  1. References should be appropriately formatted.

References are now appropriately formatted.

Round 2

Reviewer 1 Report

The authors made suggested changes.

Reviewer 2 Report

authors have responded point by point to reviewer . I believe the manuscript has been significantly improved and now warrants publication in Cells

Reviewer 3 Report

Authors performed some efforts to present their data in a convincent manner. Overall the scientific soundness is average.